# Highly Robust Ti Adhesion Layer during Terminal Reaction in Micro-Bumps

**DOI:** 10.3390/ma15124297

**Published:** 2022-06-17

**Authors:** Chen-Wei Kao, Po-Yu Kung, Chih-Chia Chang, Wei-Chen Huang, Fu-Ling Chang, C. R. Kao

**Affiliations:** 1Department of Materials Science and Engineering, National Taiwan University, Taipei 10617, Taiwan; r09527a09@ntu.edu.tw (C.-W.K.); a0987700171@gmail.com (P.-Y.K.); rroo1115@gmail.com (C.-C.C.); f08527014@ntu.edu.tw (W.-C.H.); f09527008@ntu.edu.tw (F.-L.C.); 2Advanced Research Center for Green Materials Science & Technology, National Taiwan University, Taipei 10617, Taiwan

**Keywords:** micro-joints, solid-state reaction, intermetallic, adhesion layer

## Abstract

The use of scaled-down micro-bumps in miniaturized consumer electronic products has led to the easy realization of full intermetallic solder bumps owing to the completion of the wetting layer. However, the direct contact of the intermetallic compounds (IMCs) with the adhesion layer may pose serious reliability concerns. In this study, the terminal reaction of the Ti adhesion layer with Cu–Sn IMCs was investigated by aging the micro-bumps at 200 °C. Although all of the micro-bumps transformed into intermetallic structures after aging, they exhibited a strong attachment to the Ti adhesion layer, which differs significantly from the Cr system where spalling of IMCs occurred during the solid-state reaction. Moreover, the difference in the diffusion rates between Cu and Sn might have induced void formation during aging. These voids progressed to the center of the bump through the depleting Cu layer. However, they neither affected the attachment between the IMCs and the adhesion layer nor reduced the strength of the bumps. In conclusion, the IMCs demonstrated better adhesive behavior with the Ti adhesion layer when compared to Cr, which has been used in previous studies.

## 1. Introduction

Micro-joints play an important role in three-dimensional integrated circuit (3D IC) technology in the realization of vertical chip stacking. However, the micro-joint volume is significantly little, nearly six orders of magnitude less than that of the conventional ball grid array joints or flip-chip joints and is a critical issue. With the miniaturization of solder joints and thickness reduction of under-bump metallurgy (UBM), it is observed that both Sn and Cu present limited amounts in micro-joints. This results in a large portion, or even the entire micro-joint, being occupied by intermetallic compounds (IMCs) after assembly [1]. Hence, this phenomenon introduces new reliability concerns.

The adhesion layer, such as Cr and Ti, is the main component of the UBM used in the electronics industry. Owing to the rapidly consumed solder joints and wetting layer, the IMCs would attach to the adhesion layer with limited Cu during the solid–liquid reaction. This condition causes spalling, which is reported in numerous studies [2,3,4,5,6,7]. Spalling of the IMCs causes a significant reduction in bonding strength and affects the reliability of electronic products. In addition, the formation of IMCs in Cu–Sn reactions with limited Sn has been reported previously [1,8,9,10,11,12]. These studies only focused on the micro-structural evolution in the early stages of interfacial reactions, which were abundant with either Cu or Sn during the experiment. However, fewer studies have investigated the same when the constraint volume of Cu and Sn is simultaneously converted into the full IMC joints in the terminal solid-state reaction. In a previous study, Tsai et al. [13] found that when the IMCs were attached to the Cr adhesion layer, Cu_6_Sn_5_ and Cu_3_Sn detached and left the substrate during the terminal solid-state reaction. In addition, this spalling process did not dominate the ripening or gravity effects of Cu_6_Sn_5_ compared to regular spalling. It was confirmed that the spalling was due to the high interfacial energy between the Cu and Sn intermetallic and Cr. Furthermore, void formation during the reaction was also observed, which was attributed to Cu diffusion. In contrast, the widespread usage of the Ti adhesion layer in the electronics industry is well-known, but no study has discussed the phenomenon when the IMCs contact the Ti adhesion layer during the solid-state reaction.

This study aims to investigate the micro-structure evolution of ultrathin Sn/Cu micro-bumps at the terminal reaction and observe the phenomenon when the IMCs directly contact the Ti layer during the solid-state reaction. In addition, the relationship between the voids and the reactive layer thickness are proposed and illustrated. Furthermore, since IMC formation and void nucleation deteriorate the reliability of micro-bumps [14,15,16,17,18,19], a die shear test was conducted to study the mechanical properties of the micro-bumps with different aging times. Finally, the correlations between the shear strength and evolution of IMCs during the aging process are discussed.

## 2. Experimental

Figure 1 presents a schematic of the dimensions of the Sn/Cu/Ti structure used in this study. The detailed procedure for sample preparation has been presented previously [13]. In brief, the samples were first fabricated by electroplating Cu and Sn, sequentially on the Ti adhesion layer, to form micro-bumps and then aged at 200 °C, which could accelerate the IMCs formation to reach the terminal stage of interfacial reaction, for 0, 24, 36, and 42 h to observe the evolution between layers.

After aging, the samples were cross-sectioned and polished using an ion-milling system (Hitachi IM4000Plus) with an Ar^+^ ion beam for subsequent analysis. These samples were examined using a scanning electron microscope (SEM, Hitachi SU5000) by BSE detector and transmission electron microscope (TEM, FEI Tecnai G2 F20) to observe the sample morphology and micro-structural evolution of the Cu–Sn IMCs. The chemical composition of the IMCs was analyzed by energy-dispersive X-ray spectrometry (EDX). The shear strengths at different aging times were determined by the die shear tests (Xyztec sigma) using a bonding tester with a 1 μm shear tool height from the substrate and a 10 μm/s shear speed. For each aging time, the corresponding shear strengths for 15 specimens were averaged. After the shear test, the fracture surfaces were analyzed using optical microscopy, SEM and EDX.

## 3. Results and Discussion

### 3.1. Micro-Structure Evolution of Sn/Cu Micro-Bumps before and after Aging at 200 °C

Figure 2a shows a backscattered image of the as-fabricated Sn/Cu micro-bump after the electroplating process. The average thicknesses of the Cu and Sn layers were 2.6 μm and 3 μm, respectively. In this stage, a thin Cu_6_Sn_5_ layer (0.63 μm) formed immediately at the Sn/Cu interface after the deposition process. The formation of the thin Cu_6_Sn_5_ layer is similar to that reported in the literature [20]. The study indicated that the Cu_6_Sn_5_ formed during or immediately after the deposition of Sn on Cu in bimetallic Cu–Sn films [20].

Figure 2b,c shows the zoomed-in images of the Sn/Cu micro-bump at different places, as represented by the red rectangle in Figure 2a. It was evident that Cu, Ti and Si were in contact with each other and no defects, such as cracks, were observed along their interfaces. However, apparent voids existed at the interface of the sputtered Cu and the electroplated Cu, as shown in Figure 2d. The formation of these voids would be discussed in Section 3.3.

Figure 3 shows the micro-structure of the Sn/Cu micro-bumps aged at 200 °C for 24 h. At this stage, Sn is completely converted into Cu_6_Sn_5_ and Cu_3_Sn. As shown in Figure 3b, since the Cu layer was still in contact with the IMC and Ti layer, it demonstrated good adhesion properties in the middle of the bump. Furthermore, because of the surface diffusion [21,22,23], the bump edge developed IMCs more rapidly than the middle. In addition, the Cu layer was completely consumed at the edge of the bump. Therefore, the IMCs touched the Ti adhesion layer effortlessly at the bump edge. As shown in Figure 3c, even when all the Cu layers were fully depleted, the right edge of the bump exhibited good adhesion between the IMCs and the Ti layer, without any obvious cracks. Similarly, the left edge of the bump shows the same phenomenon. This result indicates that no spalling effect occurred between the IMCs and the Ti layer, implying that the IMCs adhered to the Ti layer more stably on mutual contact.

Figure 4 shows the micro-structure of the Sn/Cu bump aged for 36 h. The Cu_3_Sn layer enlarged because of the reaction between Cu and Cu_6_Sn_5_. At this stage, as most of the Cu converted into Cu_3_Sn, the Cu layer became discontinuous and formed a series of isolated regions. Compared to the result of aging for 24 h, the interface of Cu_3_Sn/Cu/Ti exhibited more voids in the middle region after aging for 36 h. As shown in Figure 4b, although many voids appeared, no continuous cracks formed. In addition, Figure 4c shows a zoomed-in image on the right edge of the bump. Although the contact area between the IMC and the Ti layer grew owing to further Cu depletion, it still exhibited good adhesion without any cracks. Therefore, spalling did not occur between the Cu_3_Sn and Ti layers when accompanied by the formation of voids.

Figure 5 shows the micro-structure of the Sn/Cu bump aged at 200 °C for 42 h. In this stage, Cu was completely consumed and reacted with Cu_6_Sn_5_ to form Cu_3_Sn and the IMCs came in direct contact with the Ti layer through all the bumps. Although many voids formed at the Cu_3_Sn/Ti interface, no continuous gaps or detachments between the IMCs and the substrate were observed. Besides, Figure 6 exhibits the TEM observation at the interface between Cu–Sn IMCs and Ti layer. After the higher resolution was examined, the full IMC structure of micro-bump still had excellent adhesion on the Ti adhesion layer. This demonstrated a significant difference from the system of the Cr adhesion layer, which spalled IMCs from the substrate at this stage. This implies that using the Ti adhesion layer succeeded in preventing the spalling of the IMC from the substrate. This phenomenon is discussed in detail in the next section.

### 3.2. Phenomenon between Cu_3_Sn/Ti Interface

Cu–Sn IMCs may spall from the Cr and Ti surfaces when the Cu film is consumed in solid–liquid reactions [2,3,4,5,6,7]. In addition, the spalling phenomenon between the IMCs and Cr adhesion layer has also been observed during the solid-state reaction, which was confirmed by the high interfacial energy between the IMCs and Cr [13]. However, in this study, the SEM cross-sectional results illustrate that the Cu_6_Sn_5_ and Cu_3_Sn compounds stably adhered to the Ti surface, as shown in Figure 7. The IMCs attached to the Ti layer did not spall in the same manner as the Cr system. It is assumed that this phenomenon might result from the lower interfacial energy between the Cu and Sn intermetallic and Ti in the solid-state reaction. The relationship between interfacial energies can be described by the following inequality:
(1)$$\gamma_{\mathrm{Cu}–\mathrm{Sn}\;\mathrm{IMC}}{+\gamma }_{\mathrm{Ti}}{\;>\;\gamma }_{\mathrm{Cu}–\mathrm{Sn}\;\mathrm{IMC}/\mathrm{Ti}}$$
where $\gamma_{\mathrm{Cu}–\mathrm{Sn}\;\mathrm{IMC}}$ is the surface free energy of the intermetallic, $\gamma_{\mathrm{Ti}}$ is the surface free energy of Ti and $\gamma_{\mathrm{Cu}–\mathrm{Sn}\;\mathrm{IMCs}/\mathrm{Ti}}$ represents the interfacial energy between the Cu and Sn intermetallic and Ti. When Cu was depleted and the IMC came in contact with the Ti adhesion layer, the interfacial energy between the Cu and Sn intermetallics and Ti was still lower than that of the individuals. Therefore, the IMC layers did not leach from the Ti adhesion layer. In other words, spontaneous spalling did not occur in the Ti system. However, further studies are required to elucidate this mechanism.

### 3.3. Correlation of Void Formation with Reactive Metal Layers

According to the above results, micro-voids are often observed at the Cu_3_Sn/Ti interface after isothermal aging tests. In addition, many micro-voids coalesce into larger voids, as shown in Figure 8. This phenomenon could strongly weaken the properties of micro-bumps, such as their electrical and mechanical properties. Therefore, these voids are key factors that threaten the reliability of electronic packages and it is important to realize the formation of these voids.

There are two common ways of void formation in the Sn/Cu system. One is a series of voids that were formed between the sputtered Cu and the electroplated Cu, as shown in Figure 2. This phenomenon is described as the introduction of several impurities into the deposited Cu film during the electroplating. Organic additives in the electroplating solution may be the main reason for the void formation [24,25,26,27]. However, the total volume of this voiding would not change with aging time [25,28]. The other method of void formation is the Kirkendall voids, which is observed at the Cu_3_Sn layer. This voiding could be related to the Kirkendall effect, which was attributed to the unequal diffusion rates of Cu and Sn in the IMC [29] and impurity segregation [30,31,32,33,34]. It is generally believed that Cu is the dominant diffusing species in Cu_3_Sn, a larger diffusion flux of Cu would occur from the Cu substrate [35,36,37]. If the vacancies, left by the diffusing-out of Cu atoms, cannot be occupied, they would gather to form new micro-voids with impurity [30,31,32,33,34]. Consequently, if the unequal diffusion rate between Cu and Sn exists, the voids would increase continuously.

From Figure 2, Figure 3 and Figure 4, it is obvious that the voids became larger with the increase in aging time. Based on previous research [38,39], the voids tended to agglomerate together and became large, since agglomeration could reduce the surface energy. However, the voids becoming larger did not symbolize the increase in voids. Therefore, the relationship between the void quantities and the aging time is required to be determined. Two methods were used to calculate the void percentage in the experiment.
(2)$$\mathrm{Porosity}\;1\;(\%)=\frac{{\mathrm{Pores}\;\mathrm{Area}\;(\mu m}^{2})}{{\mathrm{Total}\;\mathrm{Micro}-\mathrm{bump}\;\mathrm{Area}\;(\mu m}^{2})}$$
(3)$$\mathrm{Porosity}\;2\;(\%)=\frac{{\mathrm{Pores}\;\mathrm{Area}\;(\mu m}^{2})}{{\mathrm{Cu}}_{3}\mathrm{Sn}\;\mathrm{Area}\;{(\mu m}^{2}){+\mathrm{Pores}\;\mathrm{Area}\;(\mu m}^{2})}$$
where Equation (3) is used to calculate the relationship between Cu_3_Sn layer and pores because the voids are all found in the Cu_3_Sn/Ti interface.

Based on Equations (2) and (3), the blue and black curves, respectively, exhibit a similar trend in Figure 9a,b. First, it is evident that the curves increased rapidly in region II (Sn depleted/Cu remaining region), which is the region where Sn was completely consumed and only the Cu remained. This phenomenon demonstrated that the voids kept growing due to the unequal diffusion flux, forming more Kirkendall voids. Second, in region III (Cu-depleted region), which is the region that Sn and Cu were all completely consumed, because the entire Cu was depleted, the porosity did not change with time. Consequently, no more Cu diffused to form voids.

Figure 9 also shows the relationship between the remaining Cu thickness (orange line) and Cu_3_Sn growth thickness (red line) versus the two porosities mentioned in Equations (2) and (3). Figure 9a shows that the porosity increased rapidly when the remaining Cu thickness decreased steeply. This strongly indicated again that the void formation was caused by the Cu diffusion, which is referred to as Kirkendall voids. In Figure 9b, although the Cu_3_Sn growth thickness did not fully correspond to the trend of the porosity, it showed a similar change with the porosity curve.

### 3.4. Shear Strength and Fracture Surface of Sn/Cu Micro-Bumps

Although the interfacial IMCs and void formation at the interface typically have a significant effect on the solder joint reliability [14,15,16,17,18,19], a die shear test was conducted to evaluate the effect of interfacial reactions of Sn/Cu micro-bumps in a solid-state reaction.

Figure 10 shows the SEM images and EDX mapping analysis of the fracture surface of the bump at various aging times. Shear tests were conducted from left to right. Based on previous studies, the Sn/Cu_6_Sn_5_ interface is the main factor affecting the fracture location in the as-fabricated sample [40]. Therefore, as shown in Figure 10a, the bump was broken at the Sn/Cu_6_Sn_5_ interface without aging. However, as shown in Figure 2, there were a series of voids and the formation of small cracks between the sputtered Cu and the electroplated Cu without aging. These defects do not affect the mechanical properties at this stage. From the SEM images, the fracture surface did not pass through these voids, so the mechanical strength was determined by the Sn/Cu_6_Sn_5_ interface.

Figure 10b–d shows the fracture surface after aging for 24–48 h and Figure 11 shows the phase percentages of the fracture surface with different aging times. To ensure the full IMC structure in micro-bumps, the sample aging after 48 h was used in the die shear test. In the analysis of the fracture surface, four main layers, Cu_6_Sn_5_, Cu_3_Sn, Cu and Ti, were exposed at different aging times. The Cu_3_Sn layer was only slightly exposed in these SEM images. Interestingly, the exposed Ti layer increased with decreasing Cu_6_Sn_5_ layer thickness as the aging time increased.

Figure 12 shows the correlation between the aging time and the shear strength. The strength was approximately 9 MPa without aging. This means that the Sn/Cu_6_Sn_5_ would be the main weak interface in the as-fabricated micro-bump. However, the shear strength steeply increased to 32 MPa after aging for 24 h and then remained constant after aging for 36 h and 48 h. During this period, the Cu_3_Sn layer thickened and was the major component of IMCs attached to the Ti layer through aging. In addition, the number of voids increased with the growth of Cu_3_Sn. However, we found that the number of voids has little effect on the strength. Thus, the shear strength was dominated by Cu_3_Sn/Ti adhesive interface but not by voids. This phenomenon is finally discussed below.

By combining the findings from the cross-sectional observation, fracture surface observation and strength measurement, a schematic summarizing the results of the fracture analysis is presented in Figure 13. After electroplating, a thin Cu_6_Sn_5_ layer was formed between the Cu and Sn layers; thus, the fracture occurred through the Sn/Cu_6_Sn_5_ interface with a very smooth fractured surface. When the aging time reached 24 h, the shear strength increased with the growth of Cu_3_Sn. In addition, according to the phenomenon mentioned above, the outer edge of the micro-bump formed some voids at the Cu_3_Sn/Ti interface. The stress concentration would be developed around these voids, which could more easily cause the crack growth. Therefore, the crack propagated along with the Cu_6_Sn_5_ layer and extended to the Cu_3_Sn/Ti interface. Furthermore, when the aging time increased to 36 and 48 h, the Cu_3_Sn layer grew increasingly with the depletion of the Cu layer, and the void formations also increased at the Cu_3_Sn/Ti interface. Thus, the cracks propagated more easily and extended to the Cu_3_Sn/Ti interface. This led to an increase in the percentage of the exposed Ti area on the fracture surface with increasing aging time. However, since the fracture surface mainly occurred at the Cu_3_Sn/Ti interface, the adhesion between Cu_3_Sn and Ti was the main factor controlling the shear strength. Therefore, the strength did not change with increasing aging time and the number of micro-voids. The micro-structural characterization suggested that the key reason for determination of the micro-bump strength was not the void formation but the types of the IMC interface. Therefore, it is important to realize the interfacial reaction and IMC formation during the miniaturization of micro-bumps to achieve better joint strength.

## 4. Conclusions

In this study, the micro-structural evolution of Sn/Cu micro-bumps during the solid-state reaction with the Ti adhesion layer at 200 °C was investigated. Based on the results, the following conclusions are drawn.

Under the as-fabricated condition, a thin layer of Cu_6_Sn_5_ was formed at the Cu and Sn interfaces. In addition, after the aging time increased from 24 to 42 h, the Cu_6_Sn_5_ and Cu_3_Sn phases primarily thickened and extended towards the substrate.With the complete consumption of Sn and Cu, voids could be found and extended to the entire bump at the Cu_3_Sn/Ti interface; however, these voids could not induce cracks or gaps.After aging for 42 h, the micro-bumps transformed into intermetallic (IMC) structures. Owing to the lower interfacial energy between the Cu and Sn IMCs and the Ti layer, they all attached well to each other and the micro-bump did not exhibit spontaneous spalling.The correlation of void formation is related to the unbalanced diffusion rate in Cu_3_Sn. Since the trend of porosity had highly corresponding with Cu thickness consumption, these voids could be indicated as the Kirkendall voids.In the die shear test analysis, the strength of the bumps was determined by the type of the IMC interface. In addition, after aging, the voids extended to the entire bump, and the fracture surface exposed more Ti area. This phenomenon could be attributed to the fact that the voids would only influence the fracture path.

Finally, compared to the Cr system, the Ti adhesion layer imparted excellent properties to the full IMC joints. Therefore, the Ti system can be used to obtain reliable micro-joints as an UBM layer, as it prevents spalling of the IMCs produced during the terminal solid-state reaction.

## Figures and Tables

**Figure 1 materials-15-04297-f001:**
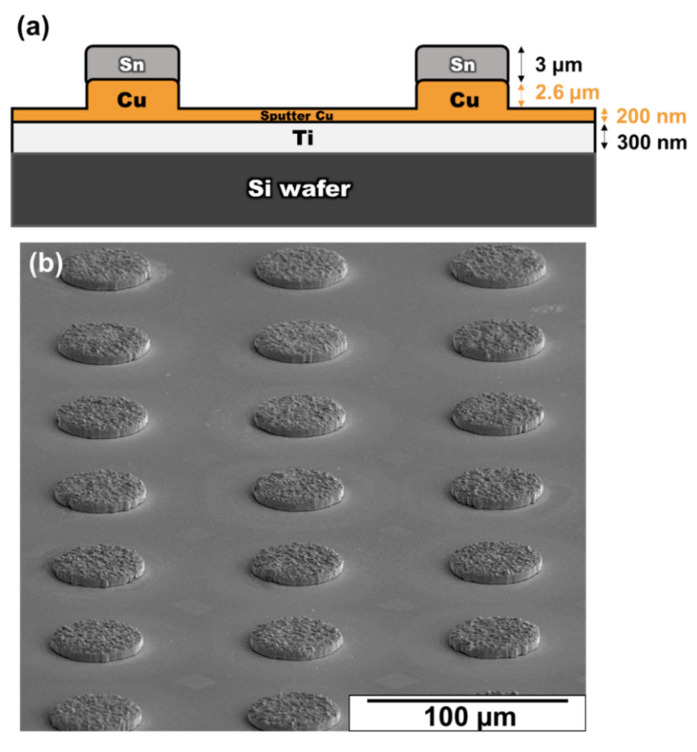
(**a**) Schematic drawing of the configuration of the Sn/Cu/Ti structure in this study; (**b**) Micrograph showing the top view of micro-bumps.

**Figure 2 materials-15-04297-f002:**
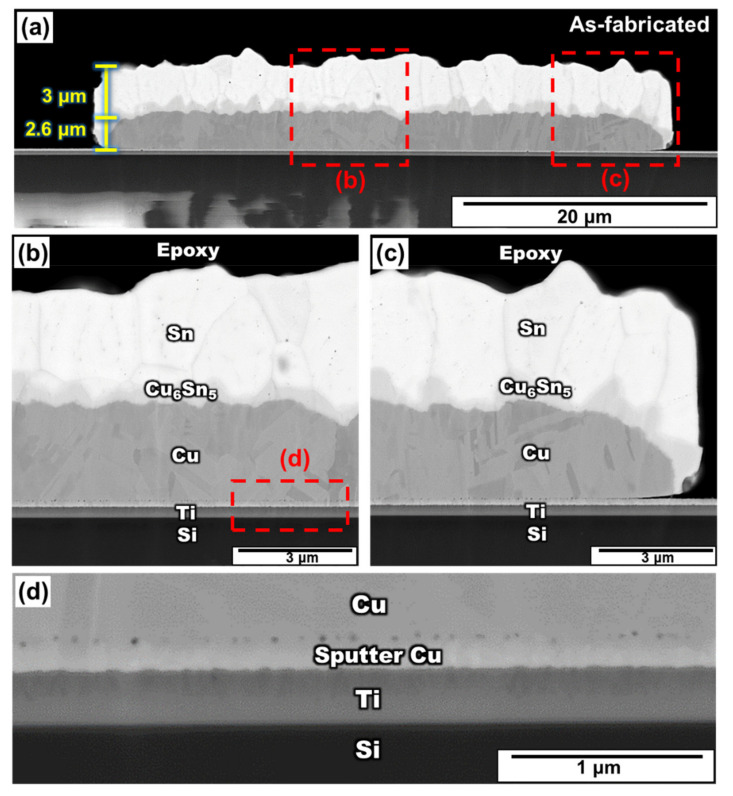
(**a**) Cross-section SEM image of the as-fabricated Sn/Cu micro-bump, and the zoomed-in image in a region near the (**b**) middle region, (**c**) right edge, and (**d**) interface between Cu and Ti layer.

**Figure 3 materials-15-04297-f003:**
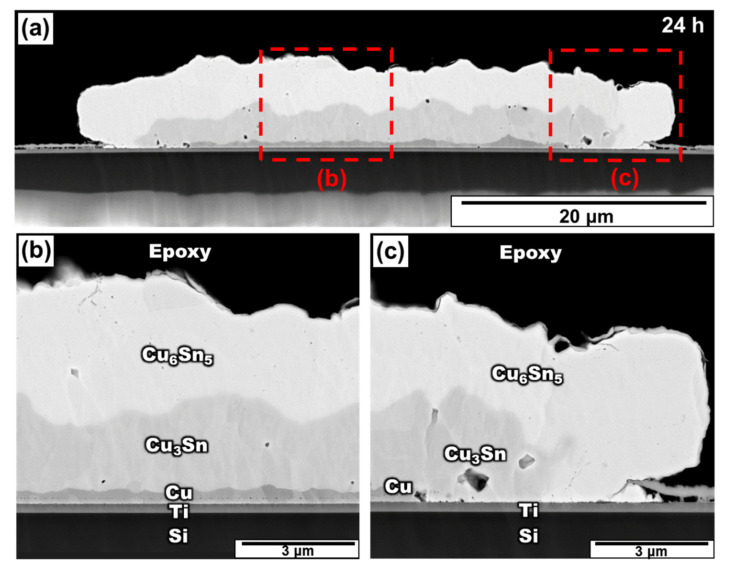
(**a**) Cross-section SEM image of the Sn/Cu micro-bump after aging at 200 °C for 24 h and the zoomed-in image in a region near the (**b**) middle region and (**c**) right edge.

**Figure 4 materials-15-04297-f004:**
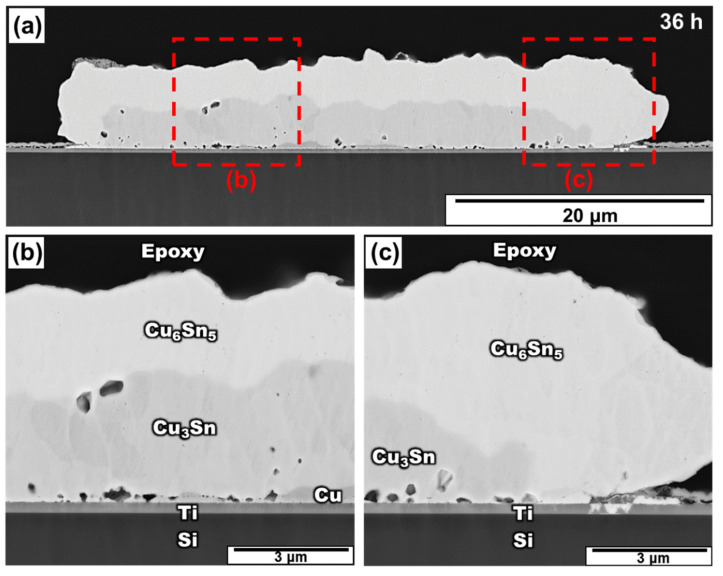
(**a**) Cross-section SEM image of the Sn/Cu micro-bump after aging at 200 °C for 36 h and the zoomed-in image in a region near the (**b**) middle region and (**c**) right edge.

**Figure 5 materials-15-04297-f005:**
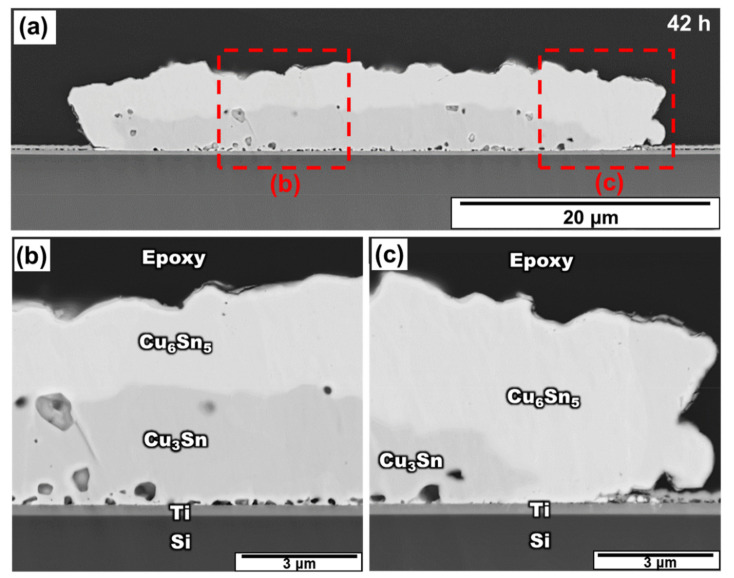
(**a**) Cross-section SEM image of the Sn/Cu micro-bump after aging at 200 °C for 42 h and the zoomed-in image in a region near the (**b**) middle region and (**c**) right edge.

**Figure 6 materials-15-04297-f006:**
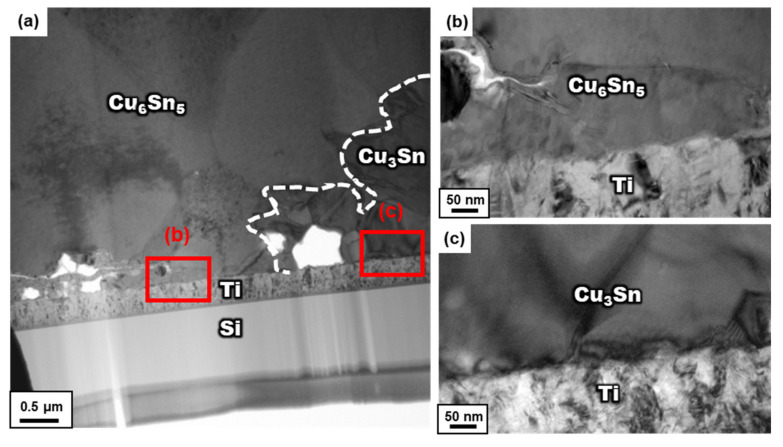
The TEM observation of the micro-bump aging after 42 h in (**a**) the region at the interface between Cu–Sn IMCs and Ti layer, and the enlarged view into the (**b**) Cu_6_Sn_5_ interface and (**c**) Cu_3_Sn interface from (**a**).

**Figure 7 materials-15-04297-f007:**
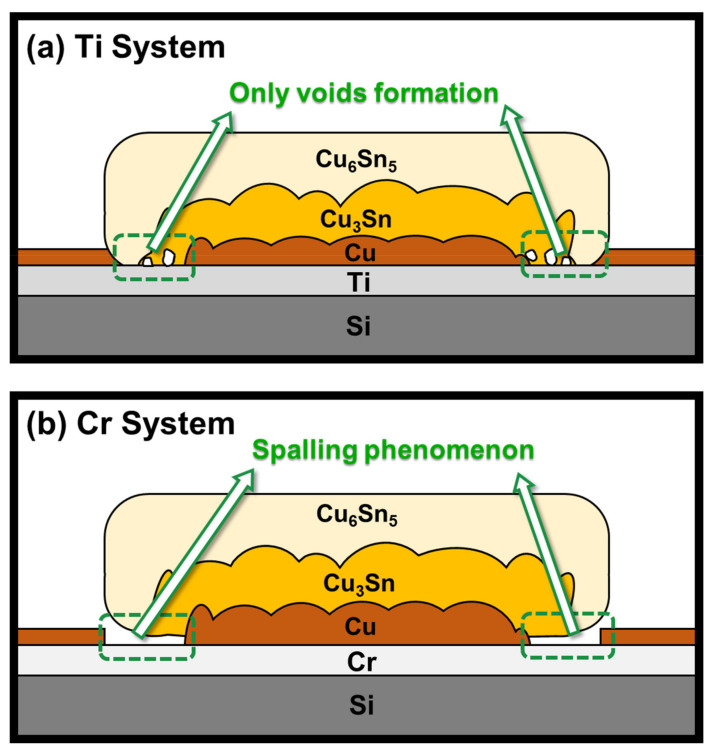
Schematic drawing of a Sn/Cu micro-bump (**a**) using the Ti adhesion layer; (**b**) using the Cr adhesion layer after aging at 200 °C with Cu existence.

**Figure 8 materials-15-04297-f008:**
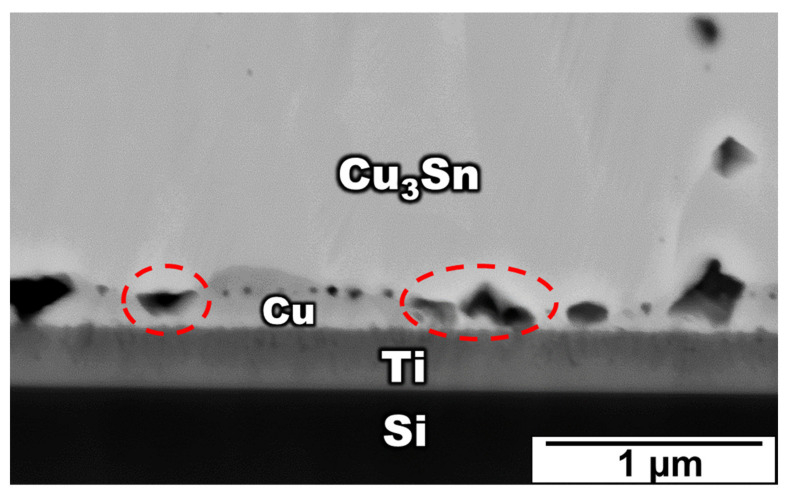
The zoomed-in image of voids formation and voids growth at the Cu_3_Sn/Cu interface after aging at 200 °C for 36 h.

**Figure 9 materials-15-04297-f009:**
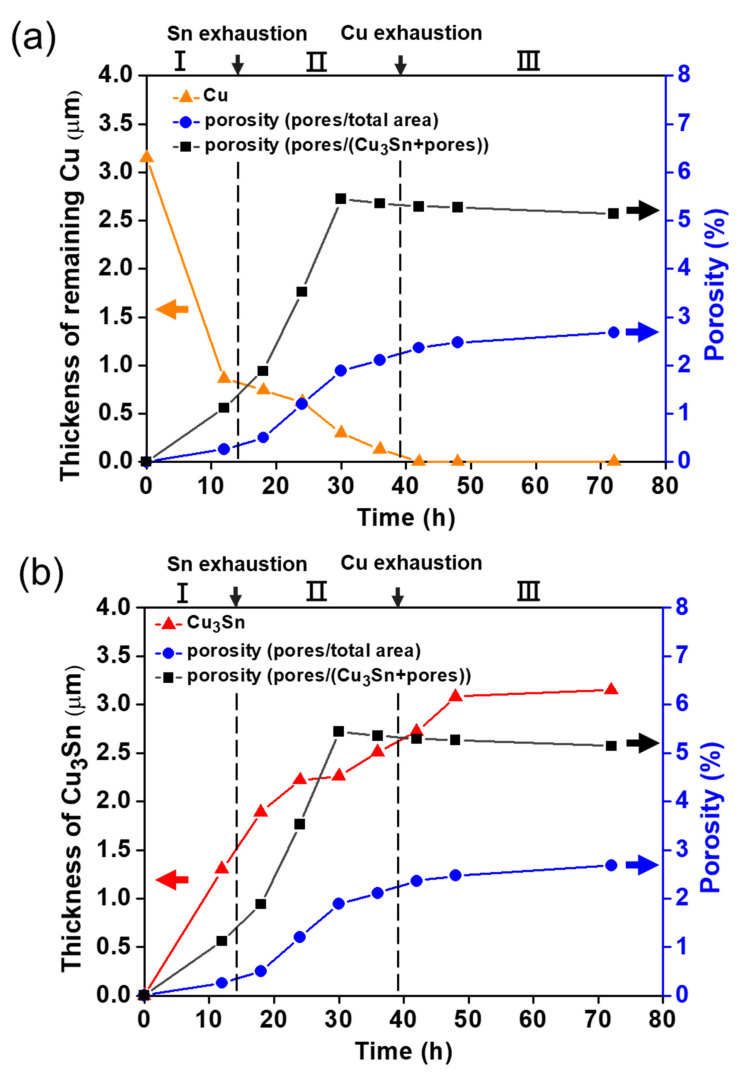
The correlation of porosity and (**a**) Cu thickness, (**b**) Cu_3_Sn thickness with different aging times.

**Figure 10 materials-15-04297-f010:**
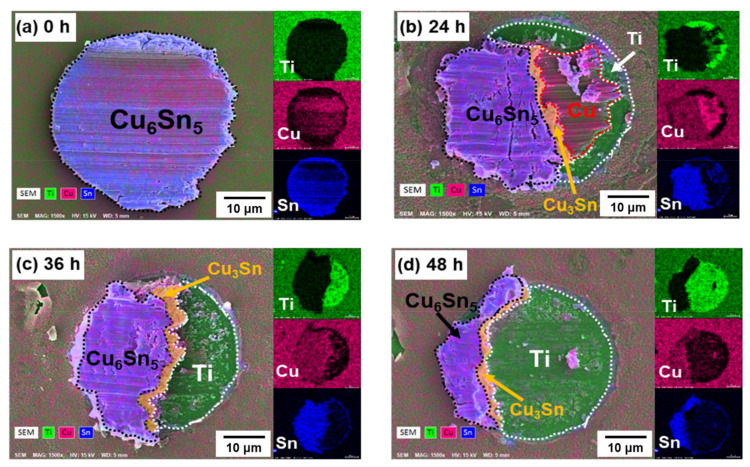
The fracture surfaces of solder joints after aging at 200 °C for: (**a**) 0 h, (**b**) 24 h, (**c**) 36 h and (**d**) 48 h with their elemental area distribution of Ti (green), Cu (red), Sn (blue).

**Figure 11 materials-15-04297-f011:**
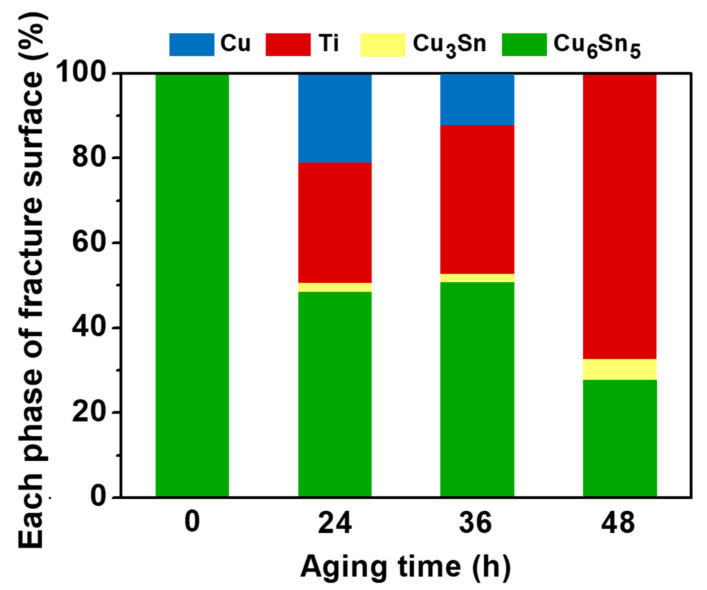
The phase percentages of fracture surface against aging time.

**Figure 12 materials-15-04297-f012:**
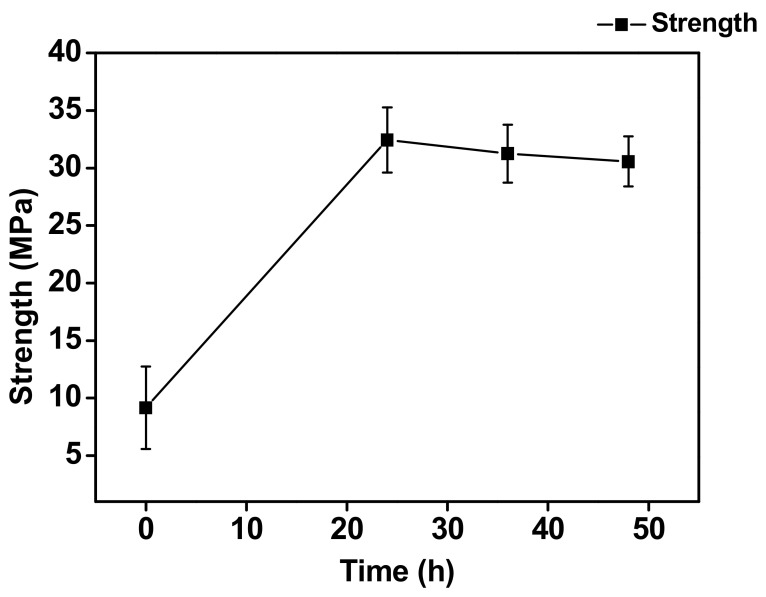
Variation of the shear strength with the aging time at 200 °C.

**Figure 13 materials-15-04297-f013:**
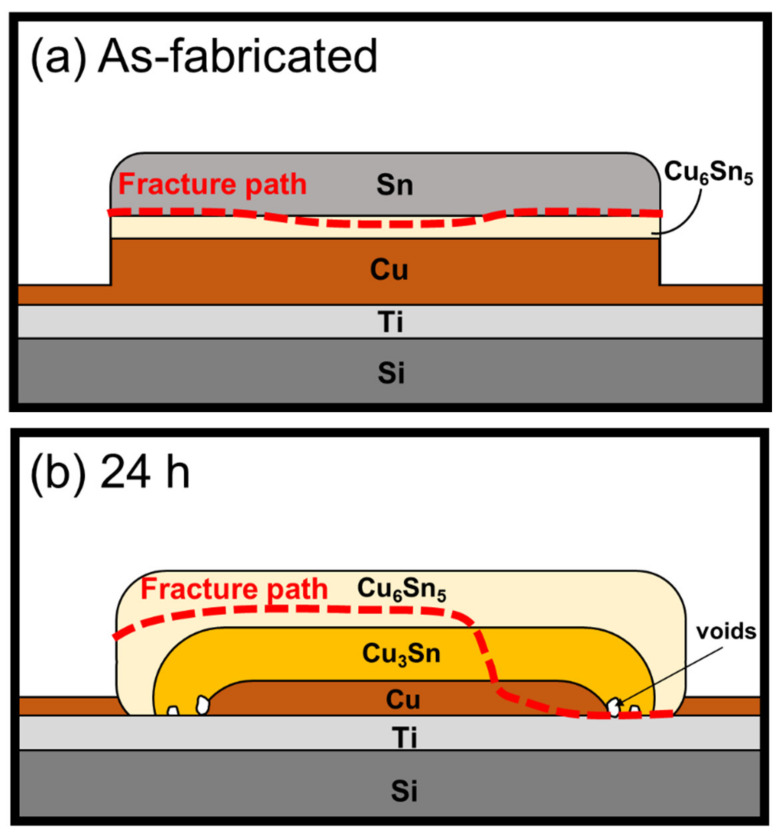
Schematic drawing showing the crack propagation of Sn/Cu micro-bumps (**a**) as-fabricated, and after (**b**) 24 h, (**c**) 36 h, and (**d**) 48 h of aging.

## Data Availability

Not applicable.

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
