# Peer review of "Highly Robust Ti Adhesion Layer during Terminal Reaction in Micro-Bumps"

_materials, 2022, doi:10.3390/ma15124297_

Round 1

Reviewer 1 Report

The article is full of important content. I enjoyed reading it. While I am recommending publication, I think that it need of several major revision that can improve quality and readability of the text:

(1) It is not clear from the text of the article why a temperature of 200 degrees was chosen to study the evolution of the microstructure? Why not investigated, for example, at 100 oC or 300 oC? Please add justification to the text of the article.

(2) Please specify in section 2 the exact time during which samples were held at 200 oC instead of 0 to 72 hours.

(3) Paragraph 3.1 describes the microstructure for samples aged at 200 oC for 0 h, 24 h, 36 h, 42 h. However, further in Figures 9-12 the results of samples obtained for 48 h are discussed. This is confusing!

(4) What method was used to determine the phases of Cu6Sn5, Cu3Sn? The paper does not contain data from X-ray phase analysis or decoding of micro-diffraction patterns obtained by transmission electron microscopy. In addition, transmission electron microscopy studies could show coherence/inconsistency of phase boundaries, which could better explain the adhesion of these phases to each other and to the substrate.

Reviewer 2 Report

From the above, I can highlight that there is a significant contribution from the interfaces within the Ti system, presumably because the Cu3Sn ions will be forced to change direction every time a new interface is encountered.

Ti system, presumably because the Cu3Sn ions will be forced to change direction each time a new interface is encountered, allowing the Cu3Sn ions to change direction each time a new interface is encountered.

will be forced to change direction every time a new interface is encountered, thus allowing for the thus allowing, thus allowing, thus allowing, thus allowing resistance, and the increase of interfaces possibly promotes the reduction of porosities.

These systems are characteristic of hard coatings, these excellent results are attributed to the effect of the interfaces.

So that it is possible to dissipate a greater amount of energy by acting as crack deflecting points?

General comment

The authors present the application of the microstructure evolution of ultrathin Sn/Cu micro bumps at the terminal reaction and observe the phenomenon when IMCs directly contact the Ti layer during the solid–state reaction, for applications where require high mechanical properties such as hardness and inert to chemical phenomenon

Review to the comments:

Abstract

In short, it is not clear the three different coatings.

To improve adhesion Ti was used?

Mentioned because as metallic sublayers, but not whether also use it to improve their anchorage to the substrate

Introduction

In the introduction, does not mention anything with respect to Microstructure evolution of micro bumps and their relationship with temperature

Experimental

The details given in the preparation of the coatings is well made and organized

What was the equipment used in the characterization of isothermal aging tests?

What was the range in which the curves were measured isotherma?

What is the model and parameters to measure the voids?

Characterization.

The characterization is done properly, like its discussion.

SEM images, the monolayers of Ti are not observed.

Reviewer 3 Report

Dear Editor,

In this manuscript, the authors studied the positive effect of the Ti adhesion layer on micro bumps during their artificial ageing at elevated temperature for various times (12-48 h).

These bumps serve as micro-joints for electronic components (chips).

It was found that the Ti layer effectively prevents spalling of the bump from the Si substrate during the ageing.

The manuscript is very well written in good English. The results are well described and adequately discussed. I really do not have any additional comments on its improvement. In my opinion, the manuscript can be accepted for publishing as it is.

Reviewer 4 Report

Both in Abstract and in Conclusions authors stated that addition of Ti produces better results, i.e. improves the properties of the IMC joints more than the addition of Cr. No explicit data on results of the Cr addition are presented, to be compared to the given data on the Ti addition effect. Some sort of comparative diagram would help to support the given statements.

Font size and type in equations should be the same as in the rest of the text.

Page 15 – lines 270-276 – the font size is larger than in the rest of the text.

Page 15 – line 292 – please add the country the Ministry of which granted the financial support to this research.
